# Maternal Differentiation of Self and Toddlers’ Sleep: The Mediating Role of Nighttime Involvement

**DOI:** 10.3390/ijerph20031714

**Published:** 2023-01-17

**Authors:** Tamar Simon, Anat Scher

**Affiliations:** Department of Counseling and Human Development, University of Haifa, Haifa 3498838, Israel

**Keywords:** differentiation of self, children, sleep, bedtime practices, nighttime parenting

## Abstract

Good sleep is essential for optimal development and adaptive functioning. Hence, identifying the factors that shape sleep quality is important. Based on the transactional model of sleep development and drawing on Bowen’s concept of differentiation of self (DoS), the present study examined the interrelations between sleep-related parental behavior, child’s sleep quality, and mothers’ DoS. A community sample of 130 mothers of 24- to 36-month-old children completed the DoS instrument and sleep questionnaires. Lower maternal DoS levels were associated with higher parental sleep-related involvement, both at bedtime and through the course of the night. Using structural equation modeling (SEM), a path analysis model indicates that maternal sleep-related involvement functions as a mediator through which the differentiation of self is related to the sleep characteristics of toddlers. As the links between parenting practices and child sleep reflect bi-directional associations, the conclusion that can be drawn from the present data is that relational aspects, such as those defined and measured by the construct of DoS, contribute to sleep–wake regulation beyond infancy. The data suggest that this construct should be considered in intervention research.

## 1. Introduction

Difficulties in initiating and maintaining continuous sleep are typical of many infants and young children, yet a concern to many parents [1]. Given the importance of good sleep for child health and development [2], identifying factors that differentiate between good and poor sleepers is important for both developmental science and childcare practice.

Guided by the transactional model of children’s sleep [3], researchers examined multiple child, parent, and contextual factors that contribute to sleep–wake regulation [4]. Focusing on the parenting domain, Sadeh, Tikotzky, and Scher [5] review parental cognitions, emotions, and behaviors that shape sleep-related interactions, and show that these factors are closely related to infant sleep. A consistent finding is that infant night waking problems are related to the higher involvement of parents at bedtime and during the night (e.g., [6,7,8,9]). It has been maintained that active parental involvement, when settling the child to sleep and in the course of the night, such as holding, feeding, rocking, etc., interferes with the infants’ ability to practice self-soothing and to develop their own sleep-related self-regulation [9,10,11].

In the realm of parent–child relationships, psychological constructs that have been linked individual differences in sleep–wake regulation include attachment security [12,13], emotional availability [14,15,16], sensitivity [17], separation anxiety [18], and cry tolerance [19]. To what extent are interpersonal and relational constructs associated with children’s sleep patterns beyond the parent–child dyad? Not many studies addressed this issue. It has been shown that parental distress marital conflicts [20], perceived social support [21], and insecure maternal attachment are risk factors for sleep problems in young children [22].

A recent systematic review [23] examined family-level constructs (outside of bedtime parenting) that contribute to infants and young children’s sleep quality. The presence of household chaos and poor quality marital relationships were directly associated with early childhood sleep problems. Longer sleep duration was associated with higher marital satisfaction and the presence of household routines. The authors recommend that future research should identify modifiable family-level factors that can be targeted, in addition to bedtime parenting, to improve sleep–wake regulation development [23].

In conceptualizing the transactional links between family-related constructs and children’s sleep, El-Sheikh and Kelly [24] highlighted moderating and mediating processes within the family that contribute to different pathways of associations between parental variables and child sleep. The authors call to expand the inquiry to domains of family functioning that have not yet been examined with respect to sleep. The present study aims to move in this direction by drawing on Bowen’s family systems theory [25,26], specifically on the psychological construct of differentiation of self.

The differentiation of self (DoS) is the cornerstone concept of Bowen’s family systems theory [25,26]. The differentiation of one’s self from the family of origin relates to a process that begins in infancy and progresses throughout childhood and adolescence to reach a basic level in early adulthood [27]. The construct of DoS focuses specifically on adults’ significant interpersonal relationships and current relations with the family of origin.

DoS refers to one’s ability to engage in a significant intimate relationship, while at the same time maintaining a tangible sense of self [28,29]. It has been argued that DoS levels reflect the overall psychological representations of relationships within the family [25,26,30] According to this view, parents with relatively high DoS maintain a solid sense of self and are less emotionally reactive, and therefore cope successfully with emotionally charged situations [25,26]. Parents with high DoS are capable of efficient emotional regulation and can maintain an emotional tie with their children while simultaneously encouraging autonomy and independence [26,31].

Studies that examined the associations between parent’s DoS and their children’s functioning showed that children of mothers with relatively high DoS presented higher resilience [32] and better emotional regulation [29,33,34,35] compared to the low-DoS group. The task set for the present study was to examine whether and how different levels of mothers’ DoS contribute to children’s sleep–wake regulation.

Compared to the large body of research on parenting and infants’ sleep (e.g., [5,36]), only a few studies have addressed toddlers’ sleep from a relational perspective (e.g., [22]). This is surprising given that the toddlerhood period posits new psychological challenges to the mother–child tie, such as the separation–individuation process [37] and the autonomy–dependence conflict [38]. Thus, investigating toddlers’ sleep from an interpersonal perspective appears to be particularly relevant, yet empirical sleep research in this domain is still scarce. The present study focused on mothers’ differentiation of self and addressed both parental sleep-related involvement and the child’s sleep characteristics. The research hypothesizes that both intense sleep-related involvement and poor sleep are linked to lower levels of mothers’ DoS.

## 2. Method

### 2.1. Participants and Procedure

We recruited 130 mothers and their 2- to 3-year-old healthy children (70 girls and 60 boys), with no identified developmental or medical problems (GPOWER analysis for a medium size effect is N = 125). The sample was recruited through free lectures held for parents in private nursery schools on aspects of toddlers’ sleep. Parents were invited by the teacher to attend the lectures, as part of the extracurricular activities taking place in the nursery schools throughout the year. About half of the toddlers’ parents participated in each lecture, of which about 90% were mothers. Among the mothers who attended the lectures, 98% (130/133) consented to take part in ‘toddlers’ sleep research’ and filled out questionnaires during a break in the lecture held for that purpose.

All mothers cohabited with the child’s father; mothers’ ages ranged from 27 to 45 years (mean age 35.5, SD = 3.8), with education durations ranging from 10 to 22 years (M = 16.5, SD = 2.4). Furthermore, 94% were born in Israel and 88% of the mothers were employed outside the home. All children attended nursery school at least five hours a day, five to six days a week. The ages of the children ranged from 24 to 36 months (mean age 29 months, SD = 3.8); 41% were firstborns.

The study was approved by the Institutional Review Board in compliance with the requirements for the ethical conduct of research with human subjects (C.N.: 075/15).

### 2.2. Instruments

The Differentiation of Self Inventory (DSI-R; [39]) is a 46-item self-report instrument that has been validated across different cultures (e.g., [27,40,41]. The inventory is a multidimensional measure consisting of four sub-scales: emotional reactivity (e.g., “People have remarked that I’m overly emotional”); emotional cutoff (e.g., “When one of my relationships becomes very intense, I feel the urge to run away from it”); fusion with others (e.g., “My self-esteem really depends on how others think of me”); and I-positions (e.g., “I tend to feel pretty stable under stress”). Each item is scored on a 6-point Likert scale; higher scores denote higher levels of differentiation of self. The mean DoS score was 3.80 (range: 2.7–4.9, SD = 0.57); Cronbach’s α of the scale was 0.90.

The ‘Infant sleep questionnaire’ (ISQ; [42]) is a short questionnaire (6-items) that describes the child’s sleep patterns and habits. The reliability and validity of the tool were established in infants and toddlers [42,43]. Based on 6 items that characterize sleep habits, including bedtime (“on average, how long does it take to settle your child off to sleep?”), night waking (“on average, how many times does your child wake each night and need resettling?”), co-sleeping (“how often do you end up taking your child into your bed because your child is upset?”), and overall maternal evaluation of sleeping difficulties (0-no problem to 4-severe problem), a total sleep problem score was provided, ranging from 0 to 35, where a high score denotes a sleep problem. In the present sample, the mean sleep problem score was 10.8 (range: 0–33, SD = 8.20); Cronbach’s α was 0.70.

Using Richman’s [44] criteria for toddlers’ sleep problems and following Scher and Asher [43], two additional indices were scored: (a) a bedtime index—the duration of settling to fall asleep and the number of nights in which there are bedtime difficulties (r = 0.70, *p* = 0.001), and (b) a nighttime index—the number of interrupted nights, the number of awakenings per night, and the average length of each awakening (Cronbach’s α = 0.71). The two indices were interrelated r = 0.34, *p* = 0.000)

Bedtime and nighttime parental interactive behaviors [8] were adapted to score questions concerning practices at bedtime (“please describe briefly how do you prepare at home for your child’s night sleep”), and as a response to the child awakening during the night (e.g., “if your child wakes up during the night, how do you react?”). Mothers were also asked to mark the method they use most to help their child fall asleep at night (“when your child is lying in bed and supposed to go to sleep, which method you use most to help him fall asleep?”). Using the categories of parental interactive sleep-related involvement described by Morrell and Cortina Borja [8], we scored maternal bedtime involvement and nighttime involvement, on a scale of one to five [1 = no involvement; 2 = the use of an object, such as a pacifier, a bottle, or a teddy bear; 3 = short involvement (2–3 min); 4 = silent passive presence next to the child; and 5 = long active involvement]. The maternal involvement scales were positively correlated with the respective sleep indices (r = 0.25, *p* = 0.010, and r = 0.42, *p* = 0.000, respectively, at bedtime and nighttime).

### 2.3. Statistical Analyses

All statistical analyses were carried out using IBM SPSS Statistics 25 for Windows version 25, with *p* level set at < 0.01.

## 3. Results

As preliminary analyses indicated that neither mothers’ age nor education level was correlated with mothers’ DoS or with sleep measures, demographic variables were not included in subsequent analyses.

### 3.1. Sleep Characteristics

Based on mothers’ reports, the mean sleep onset time was 8:36 p.m. (SD = 38 min) and the mean morning waking up time was 6:49 a.m. (SD = 36 min). The mean sleep duration was 10 h and 13 min (SD = 43 min). According to mothers’ evaluation, as measured by ISQ, 28% of the children presented a mild sleep problem and 17% presented a moderate-to-severe sleep problem. Almost all mothers (96%) reported that they help their child fall asleep at bedtime: 30% assisted the child by using an object, such as a pacifier, a bottle, or a teddy bear; 15% reported a short interaction, including talking, stroking, hugging, or singing to the child for less than five min; 17% helped the child by staying nearby silently; and 34% resorted a long interaction with the child, including talking, stroking, feeding, holding, or singing to the child for over five min. Only 4% of the mothers reported that they let the child fall asleep with no intervention. Similarly, only 3% of the mothers reported no intervention during the night. The remaining 97% of the mothers reported that they help their child fall asleep during the night: 97% of the mothers said that they use an object, such as a pacifier, a bottle, or a teddy bear; 22% reported a short interaction, also including diaper changing or escorting the child to the toilet; 6% claimed to help their child by staying nearby silently; and 32% reported long interactions, including talking, stroking, feeding, holding, or singing to the child for over five minutes.

A series of t-tests indicated that neither gender nor birth order were significantly related to mothers’ involvement, both at bedtime and during the night: (bedtime: t_(128)_ = 1.29, *p* = 0.20, t_(128)_ = 0.39, *p* = 0.69; nighttime: t_(128)_ = 1.84, *p* = 0.07; t_(128)_ = −0.28, *p* = 0.78).

### 3.2. Maternal DoS and Toddlers Sleep

As a first step, we divided the sample using the DoS quartiles and compared the sleep-related involvement of Q1 and Q3 groups. Among the mothers with the lowest DoS, 41% were characterized by high involvement compared to 16% of the mothers with highest DoS group (χ^2^_(3/130)_ = 12.97, *p* = 0.005).

A MANOVA with DoS as the independent variable and sleep measures as the dependent variable showed significant differences in sleep characteristics (F_(4,125)_ = 3.65, *p* = 0.008, ηp^2^ = 0.11). A one-way analysis of variance revealed that the difference stemmed from mothers’ sleep-related involvement, both at bedtime and during the night.

Finally, path analysis, using structural equation modeling (SEM: Amos, Version 21) and maximum likelihood estimation, produced the following indices: N = 130, Χ^2^_(1)_ = 2.95, *p* = 0.86, CFI = 0.98, NFI = 0.97, IFI = 0.98, RMSEA = 0.123, and SRMR = 0.03. The mediation model is presented in Figure 1. As the RMSEA was higher than 0.07 (0.123), we ran another model without direct paths between DoS and the sleep variables. The second model produced similar β values, with better fit indices: N = 130, Χ^2^_(3)_ = 3.44, *p* = 0.329, CFI = 0.99, NFI = 0.97, IFI = 0.99, RMSEA = 0.03, and SRMR = 0.03.

As shown in Figure 1, DoS and involvement were negatively correlated (bedtime: β = −0.24, *p* = 0.006, nighttime: β = −0.29, *p* = 0.000). Maternal bedtime involvement predicted higher reports of bedtime difficulties (β = 0.31, *p* = 0.000), but was not a significant predictor of night waking (β = 0.13, *p* = 0.13). Maternal nighttime involvement predicted the number and duration of wake episodes (β = 0.37, *p* = 0.000). To summarize, maternal bedtime and nighttime involvement serve as mediating variables through which maternal DoS indirectly influences child sleep quality.

## 4. Discussion

Overall, sleep schedules and habits of the toddlers in this study were comparable to community samples presented in other reports (e.g., [45,46]). The low-risk nature of the sample was also reflected in the DoS data, which resemble data obtained in other community samples (e.g., [47,48].

In support of our hypothesis, lower levels of maternal DoS were associated with higher maternal sleep-related involvement. This finding in accordance with the relationships perspective on child’s sleep [49], expands its scope to include the construct of differentiation of self [25], and adds to the growing body of literature on within-person variables that link parenting practices to child’s sleep [50].

The present findings show that, among toddlers, in the age range of 24 to 36 months, more disrupted sleep was associated with more involved sleep-related practices. This result adds to the accumulated evidence on the link between parent’s sleep-related involvement and sleep–wake regulation beyond infancy (e.g., [8,21,51,52]), across different cultures (e.g., [7,53,54].

Following the search for multiple paths that link parenting, family system, and child sleep [23,24], our study incorporated a family system perspective that highlights the psychological construct of differentiation of self (e.g., from family of origin) and showed that maternal sleep-related involvement functions as a mediating variable through which maternal DoS levels indirectly influence the sleep characteristics of children aged 2 to 3 years. As lower levels of maternal DoS indicate difficulty in balancing mothers’ own emotional needs with those of their children (e.g., [34]) in the context of sleep, this difficulty may entail bedtime involvement which is insensitive to the child’s self-regulation needs [49].

In showing that mothers’ sleep-related involvement serves as a mediator linking low DoS with poor sleep–wake regulation, our study highlights an interpersonal factor, within the family system, that can be targeted in order to modulate bedtime parenting and in turn to improve sleep–wake regulation [23]. Moreover, the present research is the first to underline the psychological construct of differentiation of self within a family system perspective [25] as a promising variable to be incorporated in a family system approach to the study of sleep–wake regulation [55]. Future sleep studies should include fathers and address parents’ DoS in different populations and age groups. Given that the tool is short and user-friendly, clinicians could consider using the DoS construct in early screening and intervention processes.

## 5. Conclusions

In a community sample, mothers’ representation of their interpersonal relationships, conceptualized in terms of the differentiation of self in the context of family relationships, is a valuable psychological construct for studying young children’s sleep–wake regulation. The research adds to the growing body of research on the influence of parenting factors and processes within the family to the development of sleep patterns in early childhood. With the unique contribution of showing that DoS is a psychological construct that illuminates mother–child sleep-related interactions, our study points to a worthwhile and accessible variable for inclusion in pediatric sleep research and intervention.

## Figures and Tables

**Figure 1 ijerph-20-01714-f001:**
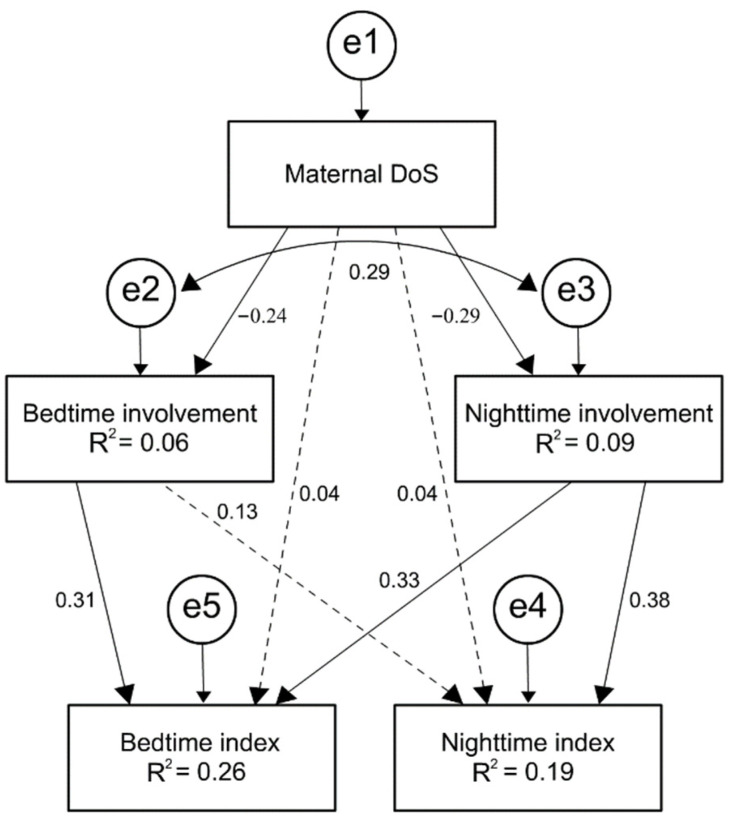
SEM model predicting child sleep difficulties based on mothers’ DoS and their sleep-related involvement. Continuous lines indicate statistical significance (<0.01).

## Data Availability

Requests to access the data set should be directed to the corresponding author.

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
