# Peer review of "Maternal Differentiation of Self and Toddlers’ Sleep: The Mediating Role of Nighttime Involvement"

_ijerph, 2023, doi:10.3390/ijerph20031714_

Round 1
Reviewer 1 Report
Dear authors this is avery interesting paper, adding novlety on the issue of beahviorla insomnia.
Few remarks:
Abstrarct: define the abbreviation SEM
"impacts the sleep characteristics of toddlers" specify better waht do you mean with impact
methods: "A higher degree of differentiation is characterized by less emotional reactivity, fusion with others, and emotional cutoff, and a greater ability to maintain positions in personal relationships": this sentemnce is not clear, define better
result: "According to mothers’ evaluation 28% of the 169 children presented a mild sleep problem, and 17% a moderate to severe sleep problem": how it is defined the sleep problem?
You should add as a limitations that you investigate only ONE paternal relation, considering the huge impact of a good relation between parents that you underlined in the introduction.
Author Response
Thank you for the points and suggestions. Here is how we incorporated them in the revised version:
- Abstract: define the abbreviation SEM
- the statistical analysis is defined
2."impacts the sleep characteristics of toddlers" specify better waht do you mean with impact
- changed to: is related
- methods: "A higher degree of differentiation is characterized by less emotional reactivity, fusion with others, and emotional cutoff, and a greater ability to maintain positions in personal relationships": this sentence is not clear, define better
- decided to omit the sentence and just specify an example for each scale
- result: "According to mothers’ evaluation 28% of the 169 children presented a mild sleep problem, and 17% a moderate to severe sleep problem": how it is defined the sleep problem?
- definition is added to the method section and mentioned in the results
- You should add as a limitations that you investigate only ONE paternal relation, considering the huge impact of a good relation between parents that you underlined in the introduction.
- In the suggestions for future studies, fathers are now mentioned.
Author Response
Thank you for your comments.
- With regard to statistical analyses, the authors reported in the text that they have conducted a SEM model. However, no latent constructs have been included, only observable variables. Thus, it would be more correct to talk about path analysis.
Indeed as the model does not include latent construct, hence we now specify that the model is based on path analysis.
- In addition to this, authors reported "maternal bedtime and nighttime involvement serve as mediating variables through which maternal DoS indirectly influences child sleep quality" (p. 5, line 210). However, no beta values and p-values related to the indirect effects have been reported. Authors must include them. The discussion is detailed and complete
The values are presented in the paragraph following the Figure